# Effects of Lactobionic Acid on Pig Growth Performance and Chemical Composition of Pork

**DOI:** 10.3390/ani12091138

**Published:** 2022-04-28

**Authors:** Jelena Zagorska, Lilija Degola, Ilvars Strazdins, Ilze Gramatina, Tatjana Kince, Ruta Galoburda

**Affiliations:** 1Faculty of Food Technology, Latvia University of Life Sciences and Technologies, Rigas Street 22, LV-3004 Jelgava, Latvia; ilze.gramatina@llu.lv (I.G.); tatjana.kince@llu.lv (T.K.); ruta.galoburda@llu.lv (R.G.); 2Faculty of Agriculture, Latvia University of Life Sciences and Technologies, Liela Street 2, LV-3001 Jelgava, Latvia; lilija.degola@llu.lv; 3LTD Latvia Dan Agro, LV-3716 Jaunberze, Latvia; lda@ldagrupa.lv

**Keywords:** pigs, live weight, slaughter yield, carcass quality, amino acids, fatty acids

## Abstract

**Simple Summary:**

Currently, due to different beneficial properties of lactobionic acid (calcium absorption, prebiotic, antioxidant, and preservative) it may find application in animal feed. However, data about its effect on animal husbandry product quality are limited. Common interest about the effect of feed supplements on pig growth performance and the chemical composition of pork is growing, which is important for farmers, meat processors and consumers. The feed supplemented with lactobionic acid promotes better utilization of feed in the digestive tract of pigs, resulting in a higher bioavailability of nutrients and a significant increase of pig live weight gain. Pork obtained from pigs fed with lactobionic acid supplemented feed had higher essential amino acid content. Fatty acid composition revealed slightly higher proportion of saturated fatty acids over unsaturated fatty acids. Further analysis of fatty acid composition indicated that the control group had better nutritional quality of fat.

**Abstract:**

Lactobionic acid is an innovative product obtained in the fermentation process of cheese or curd whey, and it has several beneficial properties. Therefore, it may have potential application in animal feeding. Currently, lactobionic acid is not widely used yet in feeding farm animals. Therefore, the aim of our study was to evaluate the effect of lactobionic acid (LBA) on pig growth performance and pork quality. Two groups of pigs (control and trial, 26 piglets each) were completed. The control group received compound feed, whereas the trial group’s feed was supplemented with LBA (0.17 kg per 100 kg of feed). Carcass weight and meat pH were determined. The subcutaneous fat layer S (mm) was measured. Lean meat content was determined using the SEUROP classification. Amino acids and fatty acids in pork were evaluated. The addition of LBA to pig feed significantly increased the live weight gain and slaughter yield of pigs, but the samples had a slightly thicker backfat layer. Results obtained showed higher concentration of amino acids in the trial group but slightly lower pork fat quality in the same group. Lactobionic acid has the potential for applications in pig feeding.

## 1. Introduction

Saving the nutritional potential of food and the effective use of resources and minimising negative influence of the processing on environment have become an urgent issues for world scientists during the last 20 years. Dairy by-product whey is a good source of sugars, minerals, and vitamins [1]; therefore, it is a valuable substrate for the production of value-added products [2]. Fermented, ammonised condensed whey may be used in small quantities as a liquid additive in pig feed. The addition of dried whey to pig feed increases the live weight gain of pigs and improves feed efficiency and the digestibility of protein and fat. Studies show that de-proteinised whey is a good animal feed additive that provides lactose and minerals [3]. Milk carbohydrates have essential roles in the intestinal development and functions of pigs [1].

Lactobionic acid (4-O-β-galactopyranosyl-D-gluconic acid) is a sugar acid, a disaccharide formed from gluconic acid and galactose. There are many studies about lactobionic acid (LBA) production; the most recent studies focus on the optimisation of biotechnological LBA production through whey lactose oxidation using various microorganisms [4,5]. Lactobionic acid possess several properties that can be successfully applied in animal feed. LBA helps to increase calcium absorption from the feed, since it can form salts with mineral cations such as calcium, potassium, sodium, and zinc. Mineral salts of lactobionic acid are also used to supplement minerals in animal feed. The complex of lactobionic acid with trace elements can be used as a feed additive for pigs, ducks, laying hens, geese, aquatic animals, and other domestic animals [6,7]. The complex of mineral elements has some advantages: lower energy consumption, low costs, environmental friendliness, and no pollution. It requires small amounts, has a significant growth-promoting effect, and has fewer side effects during use, and it can be used for a long period of time [8]. LBA is also a new strategy for calcium production. Calcium lactobionate is not so much a source of calcium as it contains less elemental (or useful) calcium, but it has a unique property that helps the body absorb more calcium from feed and supplements. This occurs by binding feed calcium ions in the stomach, intestines, and blood and helps in supply calcium to the body’s organs where calcium is most needed. The solubility of this calcium form is sixty-five-times higher than that of other calcium forms such as citrate, which is considered one of the most bioavailable. Previous studies have shown positive effects of LBA on laying hens, promoting egg shell strength [6].

Another argument for LBA’s potential application in animal feeding is its antibacterial activity, resulting in a reduced microbiological contamination of feed and possible positive effect on the animal health status. However, the most significant LBA feature is its prebiotic function, which can be applied in pig feeding, supporting the treatment of bacterial intestinal infections in monogastric animals [8]. Lactobionic acid is metabolised in the small intestine, and it is a good medium for intestinal microbiota; therefore, it is also considered to be a prebiotic that promotes desirable intestinal bacteria growth so that they can compete properly with other less desirable bacteria and pathogens, thus promoting optimal intestinal health in pigs, which in turn accelerates pig growth and development and improves meat quality [8].

Scientific literature provides research results on the development and optimisation of LBA production technologies, whereas there is only limited number of studies associated to food and feed applications of LBA [9]. The addition of 0.5–5.0% LBA to laying hen feed demonstrated eggshell reinforcing effect [7]. One of the major arguments against LBA use in food is its chelator role in human tissues during transplants [9]. Nevertheless, the use calcium lactobionate as firming agent is approved by FDA. The European Union still does not approve its use in food. According to Cardoso et al. [9], human risk evaluation studies are costly and time consuming; therefore, those would be undertaken only in case if there is clear potential observed in certain chemicals.

Lactobionic acid is an innovative product; the application of LBA in animal feeding is limited in the world due to the lack of research studies about its influence on animal health and meat quality. Therefore, the aim of our study was to evaluate the effect of lactobionic acid on pig growth performance and pork quality.

## 2. Materials and Methods

### 2.1. Animal Rearing and Performance Measurements

The study was organised in 2021 on a commercial farm “Avoti” located in Zemgale region, Latvia. For the study, two groups of pigs (control and trial) were completed, each having 26 piglets with an initial live weight of approximately 36.0 ± 0.4 kg. The animals were grouped according to pedigree, live weight, and sex. The fattening pigs in the study were Latvian Landrace × Yorkshire crossbreed pigs (M1) crossed with Durock (DJ) breed boars. The piglets in the control group received compound feed, while the trial group received feed supplemented with LBA; each group received feed in the one-colony house. The LBA solution (lactose 15.13%, protein 3.74%, fat 0.06%, LBA 11.30 g/L, dry matter 32.48%) was produced in LTD “Jaunpils pienotava” (Latvia) through cheese whey lactose fermentation by *Pseudomonas taetrolens* (DSMZ, Braunschweig, Germany). The LBA solution was included in the diet of the trial group from 7% (LBA concentration 0.11 kg per 100 kg of feed) at the beginning of the study to 15% (LBA concentration 0.23 kg per 100 kg of feed) at the end of fattening, increasing proportion by 1% of LBA solution every two weeks. Since it was the first pig feeding experiment, the average LBA (0.17 kg per 100 kg of feed) concentration was selected at a lower lever than it was indicated in the previous studies (0.5–5.0%) [9,10] and did not exceed 5% of feed amount. The chemical composition of feed in the control and trial groups were the same, and it is shown in Table 1. The feed (wheat, rye, barley, sunflower, and soybean crackers) was prepared taking into account the nutrients required by the pigs. The feeding of pigs for both groups lasted 64 days. The feed conversion ratio was calculated as proportion between feed intake per day (kg) and daily weight gain (kg) [11].

During the study, piglets’ live weight was monitored regularly by weighing. At the end of the study, all pigs were slaughtered in a commercial slaughterhouse, and pig carcasses were cooled down (4 ± 1 °C) for 24 h; after chilling, carcass weight and meat pH were determined. The pH was determined according to ISO 2917: 2004 using Jenway 3520 pH meter (Jenway, Stone, Staffordshire, UK). The subcutaneous fat (backfat thickness) layer S (mm) was measured at 6 cm off the mid-line of the carcass on the left side of the last rib with the Introscope Optical Probe (SFK, Kolding, Denmark). The percentage of lean meat was calculated according to Formula (1), and the lean meat content was determined using the SEUROP classification.
66.6708 − 0.3493 × subcutaneous fat layer (mm)(1)

The slaughter yield was calculated dividing the carcass weight by the live weight before slaughter and is expressed as a percentage [12].

### 2.2. Analysis of Pork Chemical Composition

Muscle and fat samples were taken from each carcass for chemical analysis. The chemical composition, amino acid, and fatty acid analyses of pork were performed on the combined samples obtained from six pigs in each group. Muscle and fat samples obtained from the dorsal long muscle (*Musculus longissimus thoracis et lumborum*) in the area of the last rib were separated (muscle and fat) and vacuum packed in the plastic bags. Meat samples were stored at 4 ± 1 °C. Samples were analysed after 24 h after slaughter. Chemical parameters of meat were determined according to the following methods: crude protein (% dry weight)—LVS EN ISO 5983-2: 2009; fat content (% dry weight)—LVS ISO 1443:1973; ash content (% dry weight)—ISO 936-1978; fatty acids (mg/100 g)—LVS CEN ISO/TS 17764-1; cholesterol (mg/100 g)—determined using method described by Chen et al. [13]; amino acids (% dry weight)—LVS EN ISO 13910-2005. The feed consumed in the study was calculated per 1 kg of live weight gain.

The amino acid composition of samples was used for the calculation of the nutritional value of meat proteins as an indicator of protein utilisation during digestion, as summarised below. The predicted protein efficiency ratio (PER) values of meat samples were calculated from their amino acid composition based on the equations described by Chavan et al. [14].
(2)PER−1=−0.684+0.456(Leu)−0.047(Pro)
(3)PER−2=−0.468+0.454(Leu)−0.105(Tyr)
(4)PER−3=−1.816+0.435(Met)+0.780(Leu)+0.211(His)−0.944(Tyr)

The proportions of essential amino acids (E) to the total amino acids (T) of the protein were calculated from Equation (4).
(5)$$E/T=\frac{\sum \mathrm{EAA}}{\sum \mathrm{TAA}}\times 100\left(\%\right)$$

The index of atherogenicity (IA) characterises the atherogenic potential of fatty acids [15]. For IA calculation, Equation (6) was used.
(6)$$\mathrm{IA}=\frac{C12:0+\left(4\times C14:0\right)+C16:0}{\sum \mathrm{UFA}}$$

The index of thrombogenicity (IT) characterises the thrombogenic potential of fatty acids; for calculation, Equation (7) was used.
(7)$$\mathrm{IT}=\frac{C14:0+C16:0+C18:0}{(0.5\times \sum \mathrm{MUFA})+(0.5\times \sum n-6\;\mathrm{PUFA})+(3\times \sum n-3\mathrm{PUFA})+\sum \frac{n-3}{n-6}}$$

The hypocholesterolemic/hypercholesterolemic (HH) ratio characterises the relationship between hypocholesterolemic fatty acid (cis-C18:1 and PUFA) and hypercholesterolemic fatty acid and was calculated with Equation (8).
(8)$$\mathrm{HH}=\frac{\mathrm{cis}-C18:1+\sum \mathrm{PUFA}}{C12:0+C14:0+C16:0}$$

The health-promoting index (HPI) in the present research was used to determine the nutritional value of dietary fat. For calculation, Equation (9) was used.
(9)$$\mathrm{HPI}=\frac{\sum \mathrm{UFA}}{C12:0+\left(4\cdot C14:0\right)+C16:0}$$

### 2.3. Statistical Analysis

Mathematical data processing was performed using IBM SPSS 23 software package (Microsoft Corporation, Redmond, WA, USA). The tables show the mean values of the traits and their standard errors. Significant differences in the mean values of the traits were determined by Student‘s *t*-test. Differences between means were determined at a significance level of α = 0.05. The polynomic correlation between live weight gain and time was calculated.

## 3. Results

The energy and nutrient needs of piglets depend on age, pedigree, live weight, and also on environmental conditions. During the study, the growth rates of piglets in both groups were similar (Table 2), although significant differences (*p* < 0.05) were observed between the increases in piglet live weight. In the course of the study, by analysing the results of weekly weighing of piglets, it was observed that in the trial group’s live weight gain during study had a positive polynomic correlation (*r* = 0.93). In the first 38 days of feed supplementation, live weight gain was negative compared to the control group, but in the following 26 days, the average live weight of pigs in a trial group was slightly higher.

Observations showed that pigs gladly ate feed with LBA. Additive composed 0.07 L per pig at the start of the study and 1.3 L at the end of the experiment. Neither positive nor negative changes were observed in pig health, and pigs were generally healthy in both groups, but separate cases of piglet drop were recorded. However, they were not related to feeding but were related to the general health conditions of the pigs. Daily feed intake during the study is shown in Table 3. The farm employs liquid feeding technology, when feed is prepared in a liquid form. One pig ate an average of the following amounts per day: a control group of 10.2 kg and a trial group of 9.54 kg. The consumption of dry feed was 2.54 and 2.47 kg per kg of live weight gain in the control and trial pig groups, respectively. Thus, for the trial group, it was by 0.07 kg less compared to the control group. This shows that the addition of LBA promotes a better utilization of feed in the digestive tract of pigs, resulting in a higher bioavailability of nutrients.

Cold pig carcass traits did not differ significantly (*p* > 0.05) between groups (Table 4). The thickness of the backfat was for 2.1 mm larger (Figure 1), but the pork classification class according to SEUROP was “S” (extra) for both groups.

Higher crude protein content in muscle tissue (2%) and cholesterol in fat was observed in pork from trail group, but the difference was not significant (*p* > 0.05) (Table 5).

The effects of feed supplementation with LBA on the amino acid composition of *M. longissimus thoracis et lumborum* are shown in Table 6. The supplementation with LBA did not significantly affect (*p* > 0.05) glycine (Gly), tyrosine (Tyr), and methionine (Met) content in meat samples. However, the concentrations of essential amino acids (EAA), including valine (Val), threonine (Thr), isoleucine (Ile), leucine (Leu), lysine (Lys), histidine (His) and phenylalanine (Phe), and non-essential amino acids, except tyrosine (Tyr), were significantly higher in pork from the trial group (*p* < 0.05) compared to the control group. Moreover, higher PER values were calculated for trial group pork samples compared to the control group. The proportion of essential amino acids to the total amino acids of the protein was higher for the trial group sample.

The fatty acid profile of the meat samples studied is presented as area percentage (%), as reported in Table 7. The results of the present research demonstrated that the fatty acid profile differed between groups in response to the diets that showed higher content of flavour amino acids, essential amino acids, flavour, and essential amino acids in trial group pork.

In the present research, two health-related lipid indices, atherogenic index (IA) and thrombogenic index (IT), were calculated. IA did not differ for the samples analysed. this means that feed supplemented with LBA did not influence the proportion between the sum of the main saturated fatty acids and that of the main classes of unsaturated in analysed meat samples. 

The IT characterises the thrombogenic potential of fatty acids. It is necessary to indicate that the consumption of foods with a lower IT is beneficial for human health [15]. In the present research, the IT index of control group samples was significantly lower (*p* < 0.05) compared to trial group pork samples.

No significant differences were found between the hypocholesterolemic/hypercholesterolemic ratios of meat samples analysed; they were very close (Table 7).

## 4. Discussion

The total fat content of pork varies widely from 1% to 15% apparently due to ingested feed or genetic factors such as the fat content of 7.24% for M1 × DJ crossbreeds and 3.23% for M1 × PJ crossbreeds. Latvian Yorkshire pigs [16], as well as pigs of local origin and their crossbreeds [17,18], also had a high fat content in *M. longissimus thoracis et lumborum*.

The pH values of muscle samples were measured 24 h after slaughter. The variation in pH values of pork could be due to post-mortem glycolysis. Coi et al. [19] reported that the final pH value of meat could also be affected by breed, feeding, environment, slaughtering, and the post-management of carcasses. 

The amount of cholesterol in pork depends on various factors. Faria et al. [20] revealed that it was around 84.75 mg/100 g in ham meat and 87.25 mg/100 g in *M. longissimus thoracis et lumborum* and had an interaction with pig sex and fat content in the diet. Cholesterol content is high in pork fat; in our study, it was between 303 and 312 mg/100 g of dry matter.

Traditionally meat is a very important source of essential amino acids in the human diet [21]. Ma et al. [22] mentioned in their review that glutamic acid characterises the flavour of pork; however, histidine, arginine, methionine, valine, tryptophan, tyrosine, isoleucine, leucine, and phenylalanine produced more bitter flavours, while alanine, serine, threonine, glycine, lysine, proline, and hydroxyproline produced sweeter flavours. In the present research, it was established that feed supplemented with LBA resulted in significantly (*p* < 0.05) higher proportions of essential amino acids from the total amino acids of the protein. The amino acid content obtained was close to the data summarised by Tian et al. [23]. Relatively, a highly predicted protein efficiency ratio (PER) value of 2.65 relative to mechanically deboned red meat was reported in study of Lee et al. [24], which is very close to data summarised in the present research and indicates a possibly high protein efficiency of meat samples analysed when used in the human diet. Such results can be explained by the higher bioavailability of nutrients in feed supplemented with LBA since the microbiota were promoted by prebiotic, namely LBA. Since the digestibility of amino acids in a digestive tract is extremely important for the bioavailability of amino acids, a growth of intestinal microbiota should be promoted. Previous studies have proved the positive effects of LBA, which are comparable to those obtained with lactulose, on *Lactobacillus paracasei* and *Lactobacillus rhamnosus* [25]. LBA has certain features of dietary fibre: It is not absorbed in the small intestine, and it is a good carbon source for intestinal microbiota. Moreover, current results explain a lower amount of feed per kg live weight gain in the trial group. LBA concentrations (0.17 kg per 100 kg of feed) applied in the current research were lower than that used in the previous studies. The results can be explained with increased prebiotic effect of LBA combined with fibre included in the main feed (see Table 1).

Data collected from the scientific literature on the chemical composition of pork proteins, total fat content, SFA, MUFA, and PUFA content, as well as on minerals that are important in human diet, show that the total protein content of pork is stable and ranged from 19 to 24% and was almost independent of the genetic background and environment of the animals [26]. This was also confirmed by a study [27] where significantly higher crude protein and lower fat contents were observed in M1 × PJ crossbred pigs (*p* < 0.05); the average crude protein content of both (M1 × PJ and M1 × DJ) genotypes in *M. longissimus thoracis et lumborum* ranged from 20.81 to 22.11%. In studies by other authors [12,17,28], the average content of crude protein in the *M. longissimus thoracis et lumborum* was similar to our results, while in several studies [29,30], the protein content in the long back muscle of pigs exceeded 23%.

The results of fatty acid composition in studies by other scientists showed that the highest content of saturated fatty acids was for palmitic acid (21–25%), but the proportion of eucosanoic acid was less than 1%. Of the monounsaturated fatty acids, palmitoleic acid levels were remarkably high and linoleic acid levels were outstanding among polyunsaturated fatty acids. The fatty acid composition of pig muscles is influenced by a number of factors, including fatness, body weight, age, energy intake, and dietary fatty acid composition. There are also factors related to gender, de novo fatty acid synthesis, and genetic background [19,31].

As it was mentioned by Carneiro et al. [32], IA and IT provide the stimulus potential of platelet aggregation; therefore, lower IA and IT values provide greater amounts of antiatherogenic fatty acids in fat or oil with greater potential to prevent coronary heart disease in the future. For the SFA series, even if saturated fatty acids are involved in atherogenic and thrombogenic processes, not all of them exhibit the same behaviour with respect to elevated serum cholesterol. From SFA, lauric acid (C12:0), myristic acid (C14:0), and palmitic acid (C16:0) increased plasma cholesterol levels. In addition, C14:0 was considered to have the most harmful cardiovascular effects on humans, almost four times the effects of C1:0 and C16:0. The group of saturated fatty acids with animal fats is dominated by palmitic acid and stearic acid [33]. However, a fatty acid composition with a lower IA and IT has better nutritional quality [15]. From present data, it can be concluded that the fatty acid composition of the control group provided better nutritional quality of pork compared to the trial group, which possibly indicates that LBA influences the fatty acid formation processes in meat since one of the key factors influencing fatty acid composition is animal feed [34]. It is necessary to note that lactose concentration was sufficient and equal to lactobionic acid concentrations in LBA additives, and it could be the factor that had influences on pork quality too. In the present research, a lower IA index was obtained in meat samples analysed compared to the values detected in Kušec et al.’s [34] study—1.260 ± 0.312. However, Chen and Liu [15] indicated that the IA index in pork (DanBred × PIC terminal line) ranged from 0.27 to 0.31. The study of Kasprzyk et al. [33] reported that IT ranged from 1.12 to 1.14, while the mean value of IA was 0.46.

The hypocholesterolemic/hypercholesterolemic ratio (HH) for meat products ranges from 1.27 to 2.786 [15], which corresponds to the data obtained in the present study 2.09 ± 0.02 and 1.99 ± 0.01 for the control and trial groups accordingly. Compared to the PUFA/SFA ratio, the HH ratio may more accurately reflect the effects of fatty acid composition on cardiovascular disease. Similarly to IA and IT, HH could include more types of fatty acids, such as other molecular types of MUFA, and different molecular fatty acid types may be assigned different weights [15].

The health-promoting index (HPI) is traditionally calculated for dairy products, and it ranges from 0.16 to 0.68. Moreover, dairy products with a high HPI value are considered to be more beneficial for human health [15]. In the present research, the health-promoting index of analysed meat samples was similar for both pork samples, and it was approximately three-times higher than in dairy products.

LBA has the potential for applications in pig feeding, but further research should be conducted by paying attention to LBA quality during storage and production, with the aim to ensure farms have high and stable LBA quality. 

## 5. Conclusions

The addition of lactobionic acid to pig feed significantly increased the live weight gain of pigs (*p* = 0.04) and decreased feed consumption per 1 kg live weight gain by 0.07 kg. The slaughter yield was higher in animals fed with diets supplemented with lactobionic acid, but they also had a slightly thicker backfat layer, which could reduce the attractiveness of the meat relative to the opinion of consumers. The obtained results showed the influence of lactobionic acid on chemical parameters of pork: a higher concentration of amino acids, but a negative effect should be mentioned for pork fat quality in the trial group. Differences in pork quality could be explained by the different nutrient digestibility and bioavailability of feeds supplemented with lactobionic acid.

## Figures and Tables

**Figure 1 animals-12-01138-f001:**
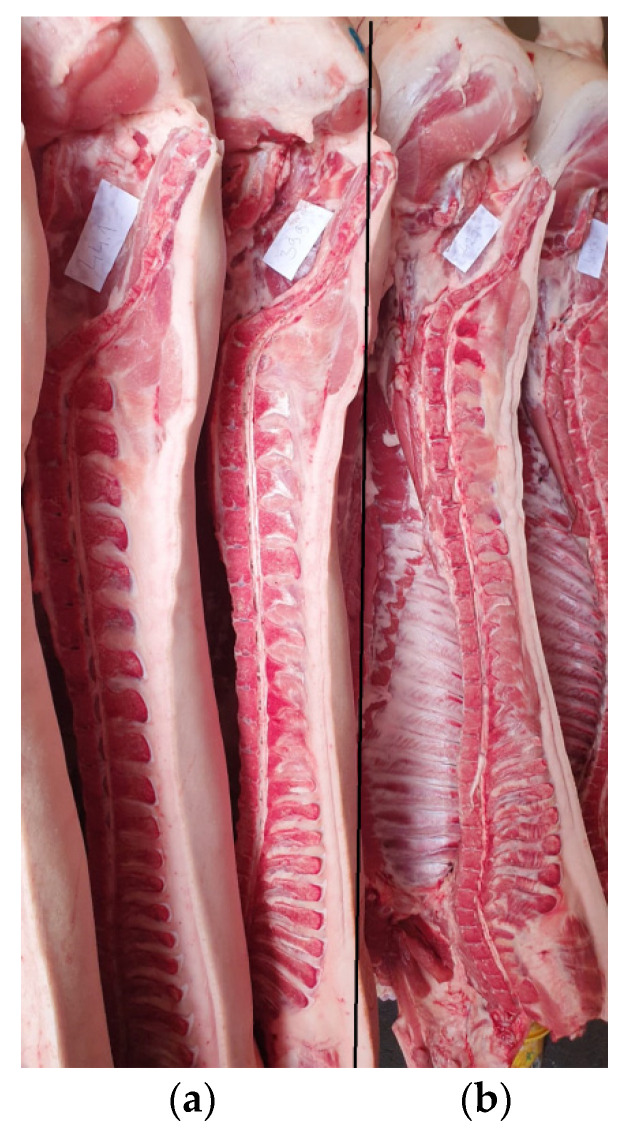
Pork from trial (**a**) and control (**b**) group pigs.

**Table 1 animals-12-01138-t001:** Chemical composition of feed (kg per 100 kg of feed).

Content	In Pig Growth Period		At the End of Fattening Period	
	Control Group	Trial Group	Control Group	Trial Group
Water	69.31	69.07	69.28	68.77
Dry matter, including:	30.69	30.93	30.72	31.23
protein	17.70	17.20	16.80	16.70
fat	2.85	2.88	2.87	2.18
fibre	4.29	4.53	5.32	5.45
ash	5.85	6.14	5.73	6.18
LBA	0.00	0.11	0.00	0.23

**Table 2 animals-12-01138-t002:** Indices of pig growth during 64 trial days.

Indices	Control Group	Trial Group	*p* Value
Live weight at beginning of trial, kg	36.30 ± 0.35	36.40 ± 0.36	0.81
Live weight at the end of trial, kg	108.90 ± 0.62 ^b^	112.70 ± 0.51 ^a^	0.04
Live weight gain, kg	72.60 ± 1.91 ^b^	76.30 ± 1.37 ^a^	0.04
Daily live weight gain, kg	1.13 ± 0.03	1.19 ± 0.02	0.32
Feed conversion ratio	2.54 ± 0.20	2.49 ± 0.18	0.96

^a,b^ Different superscript letters in the same row denote significant differences (*p* < 0.05). Values are arithmetical means (*n* = 25 per group) ± standard error.

**Table 3 animals-12-01138-t003:** Average daily feed intake per pig.

Traits	Control Group	Trial Group	*p* Value
Dry matter, kg	2.87 ± 0.08 ^b^	2.96 ± 0.09 ^a^	0.02
Water, L	7.33 ± 0.22 ^a^	6.58 ± 0.20 ^b^	0.01
Lactobionic acid solution, L	0.00 ± 0.00	1.13 ± 0.01	-

^a,b^ Different superscript letters in the same row denote significant differences (*p* < 0.05).Values are arithmetical means (*n* = 25 per group) ± standard error.

**Table 4 animals-12-01138-t004:** Pig carcass traits.

Traits	Control Group	Trial Group	*p* Value
Live weight before slaughtering, kg	104.70 ± 6.20	109.10 ± 7.10	0.58
Carcass weight, kg	74.80 ± 4.45	77.20 ± 5.49	0.17
Slaughter yield, %	71.50 ± 1.30	70.80 ± 2.70	0.70
Backfat thickness, mm	13.60 ± 2.87	15.70 ± 4.30	0.71
Lean meat content, %	61. 92 ± 3.10	61.19 ± 3.43	0.41
Pork pH	5.50 ± 0.03	5.50 ± 0.03	0.26

Values are arithmetical means (*n* = 25 per group) ± standard error.

**Table 5 animals-12-01138-t005:** Content of crude protein, fat, ash in muscle, and cholesterol in fat (in dry matter).

Traits	Control Group	Trial Group	*p* Value
Crude protein content in muscle, %	83.54 ± 3.81	85.39 ± 2.76	0.06
Fat content in muscle, %	8.30 ± 0.43 ^a^	7.00 ± 0.37 ^b^	0.01
Ash content, %	6.15 ± 0.22 ^a^	5.60 ± 0.15 ^b^	0.01
Cholesterol content in fat, %	0.30 ± 0.01	0.31 ± 0.01	0.40

^a,b^ Different superscript letters in the same row denote significant differences (*p* < 0.05).

**Table 6 animals-12-01138-t006:** Amino acid profile of *M. longissimus thoracis et lumborum* according to pork group.

Items	Control Group	Trial Group	*p* Value
Flavour amino acids, % of dry weight			
Aspartic acid (Asp)	7.34 ± 0.01	7.58 ± 0.05	0.07
Proline (Pro)	3.45 ± 0.03 ^b^	3.70 ± 0.02 ^a^	0.01
Arginine (Arg)	4.88 ± 0.01 ^b^	5.11 ± 0.01 ^a^	0.02
Serine (Ser)	2.88 ± 0.02 ^b^	2.98 ± 0.01 ^a^	0.05
Glutamic acid (Glu)	12.05 ± 0.04 ^b^	12.32 ± 0.03 ^a^	0.01
Glycine (Gly)	3.52 ± 0.01	3.54 ± 0.02	0.38
Alanine (Ala)	4.28 ± 0.02 ^b^	4.43 ± 0.01 ^a^	0.01
Cysteine (Cys)	1.05 ± 0.01 ^b^	1.19 ± 0.04 ^a^	0.01
Essential amino acids, % of dry weight			
Valine (Val)	3.70 ± 0.01 ^b^	4.00 ± 0.03 ^a^	0.01
Threonine (Thr)	3.29 ± 0.02 ^b^	3.48 ± 0.01 ^a^	0.01
Isoleucine (Ile)	3.68 ± 0.02 ^b^	3.91 ± 0.02 ^a^	0.02
Leucine (Leu)	6.20 ± 0.01 ^b^	6.37 ± 0.01 ^a^	0.02
Tyrosine (Tyr)	2.80 ± 0.04	2.86 ± 0.02	0.26
Lysine (Lys)	7.37 ± 0.02 ^b^	7.57 ± 0.06 ^a^	0.03
Histidine (His)	3.41 ± 0.01 ^b^	3.84 ± 0.01 ^a^	0.01
Flavour amino acid and essential amino acids, % of dry weight			
Phenylalanine (Phe)	3.01 ± 0.02 ^b^	3.19 ± 0.03 ^a^	0.05
Methionine (Met)	2.64 ± 0.01 ^b^	2.70 ± 0.01 ^a^	0.03
Total	75.56	78.77	-
Amino acid quality indices			
PER-1	2.03	2.08	-
PER-2	2.05	2.12	-
PER-3	2.24	2.43	-
E/T, %	47.78	48.14	-

^a,b^ Different superscript letters in the same row denote significant differences (*p* < 0.05). Data are reported as means ± standard deviation, *n* = 3 per dietary treatment).

**Table 7 animals-12-01138-t007:** Fatty acid profile (% dry weight) of *M. longissimus thoracis et lumborum*.

Fatty Acid	Control Group	Trial Group	*p* Value
11:0	ND	ND	-
12:0	1.11 ± 0.23 ^a^	0.77 ± 0.01 ^b^	0.02
13:0	1.54 ± 0.02	1.52 ± 0.03	0.06
14:0	0.31 ± 0.26	0.31 ± 0.16	0.97
14:1	ND	ND	-
15:0	1.37 ± 0.01 ^a^	1.35 ± 0.02 ^b^	0.05
15:1	ND	ND	-
16:0	23.66 ± 0.46	24.05 ± 0.75	0.55
16:1 n-7c	0.56 ± 0.01	0.53 ±0.13	0.06
17:0	0.67 ± 0.04	0.63 ± 0.02	0.07
17:1	0.38 ± 0.17 ^b^	0.50 ± 0.01 ^a^	0.05
18:0	12.36 ± 0.39	12.60 ± 0.42	0.52
18:1 n-9t	ND	ND	-
18:1 n-9c	42.47 ± 0.62 ^a^	41.32 ± 0.78 ^b^	0.02
18:1 n-7t	1.63 ± 0.11 ^b^	3.53 ± 0.05 ^a^	0.01
18:2 n-6t	ND	ND	-
18:2 n-6c	5.82 ± 0.06	ND	-
18:3 n-6	0.40 ± 0.01	ND	-
18:3 n-3	0.48 ± 0.10	ND	-
20:0	0.24 ± 0.07	ND	-
18:2	0.24 ± 0.01 ^b^	6.32 ± 0.11 ^a^	0.01
18:2	0.35 ± 0.07 ^b^	0.53 ± 0.05 ^a^	0.02
20:1 n-9c	1.33 ± 0.17 ^a^	0.47 ± 0.03 ^b^	0.01
21:0	ND	0.53 ± 0.15	-
20:2 n-6c	0.41 ± 0.01	0.32 ± 0.02	0.29
20:3 n-6c	0.44 ± 0.08 ^a^	0.33 ± 0.01 ^b^	0.01
20:4 n-6c	ND	1.63 ± 0.05	-
20:3 n-3c	ND	ND	-
22:0	1.39 ± 0.06	0.43 ± 0.03	0.58
22:1 n-9c	0.29 ± 0.03	ND	-
20:5 n-3c	0.76 ± 0.06	ND	-
23:0	ND	0.19 ± 0.01	-
22:2 n-6	0.43 ± 0.13	ND	-
24:0	ND	ND	-
24:1 n-9c	0.62 ± 0.16 ^a^	0.12 ± 0.01 ^b^	0.05
22:6 n-3c	0.73 ± 0.03	ND	-
Σ SFA	42.67 ± 1.53	43.13 ± 1.67	0.07
Σ MUFA	47.28 ± 1.27	48.03 ± 1.23	0.06
Σ PUFA	10.06 ± 0.53 ^a^	8.84 ± 0.26 ^b^	0.05
SFA/UFA	0.74 ± 0.01	0.76 ± 0.01	0.06
MUFA/PUFA	4.70 ± 0.01 ^b^	5.43 ± 0.01 ^a^	0.01
n3/n6	0.11 ± 0.01 ^b^	0.26 ± 0.01 ^a^	0.01
IA	0.46 ± 0.03	0.46 ± 0.01	0.10
IT	1.09 ± 0.01 ^b^	1.23 ± 0.02 ^a^	0.05
HH	2.09 ± 0.02 ^a^	1.99 ± 0.01^b^	0.01
HPI	2.20 ± 0.01	2.18 ± 0.01	0.06

^a,b^ Different superscript letters in the same row denote significant differences (*p* < 0.05). Data are reported as means ± standard deviation (*n* = 3 per dietary treatment). ND—below detection limit.

## Data Availability

Data are available from authors upon reasonable request.

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
