# Peer review of "Effects of Lactobionic Acid on Pig Growth Performance and Chemical Composition of Pork"

_animals, 2022, doi:10.3390/ani12091138_

Round 1

Reviewer 1 Report

Check spelling and grammar (i.e. use comma after however in line 12)

In line 55-68 (can lactobionic acid be absorbed in the small intestine); theoretically, there is no way to absorb disaccharides since all CHO must be broken down into monosaccharides before being absorbed (please make sure your introduction is correct)

LBA solution was added 7%-15% and concentration of 0.36% In dry matter. Can you elaborate on it? it is confusing (lines 93-95)

Ca you use protein efficiency ratio to measure nutritional value of pork’s meat? PER is used to measure efficiency of protein utilization in terms of animal physiology (lines 125-133)

For statistical analysis, please make sure to mention the type of statistical analysis you conducted (was it ANOVA? GLM? T-test?) (lines 148-152)

Table 2. Make sure that the numbers are aligned. Eliminate the row showing trial days and explain the duration of the trial in the heading of the table

Table 3. Where are the p-values?

Table 5. Where are the p-values?

For all tables: use the same amount of decimals

Elaborate more in the “symbiotic effect of LBA and fiber” (lines 260-261). Make sure to use statistical interactions if you mention this fact. It needs to be backed up by your data

Reword the first sentence of the conclusions (lines 319-321). you said there is increase live weight (p=0.04) and feed consumption per 1 kg live weight was by 0.07 kg lower à what do you really mean with that?

Author Response

Dear Reviewer,

we want to thank you for your very helpful and constructive feedback reviewing our manuscript “Effects of Lactobionic Acid on Pig Growth Performance and Chemical Composition of Pork” for the journal “Animals”, for your valuable comments and recommendations, for your time. We addressed all the questions raised and made corrections accordingly. The response to each question raised is given in the document attached.

Sincerely Yours,

Jelena Zagorska

Reviewer 2 Report

Line 18-19 What’s your mean “fat quality”, the fat content, fatty acids composition or antioxidant capacity…?

Line 25 The supplement level of LBA should be indicated here.

Line 31-32 The sentence “Differences in pork quality could be…by LBA” was improper as part of the conclusion. This expression has no data support in the present study.

Line 86 The body weight data should supplement the SEM value.

Line 86-87 Animals in the same group were feed in one colony house or individual pens?

Line 103 Carcass weight and pH were determinated immediately? No Cooling flushing process? If so, the expression of carcass weight must change to “hot carcass weight”. In addition, the pH value of 24 h after slaughter also should be supplemented.

Line 149 Information about company, state, country of the software should be supplemented.

Line 153 Please calculate the feed conversion ratio and list it in the table.

Line 157-158 Analysis the data of weekly weighting should consider the time effect. So the analysis method of these data should supplemented in the statistical analysis part.

Line 160 There is no need use the “*” to mark p value.

Line 175 Standard error of the dry matter and water intake data were lost.

Line 180 The same question about carcass weight, hot cold carcass weight?

Line 185 Fat content was obviously higher in the control group than that in the trail group from figure 1, but the backfat thickness value was lower in the control group . Please explain the potential reasons.

Line 175, 188, 200 and 220 P value must supplemented in table 3, 5, 6, 7 as expressed in table 2.

Line 188 Could you provide the fat, dry matter and ash content in the muscle?

Line 220 In table 7, please calculate the value of SFA/UFA, and MUFA/PUFA, n3/n6.

Author Response

(The authors gave the same response as above.)

Round 2

Reviewer 2 Report

For all the tables,  superscript letters were no necessary when the P value greater than 0.05.

Author Response

Dear Reviewer,

Thanks for your time and efforts considering our manuscript. After your advice, we decided to delete superscript letters from the tables, because those duplicate the same information which is clearly shown by p-values.

Sincerely Yours,

Jelena Zagorska
